# Exploring the Maintaining Period and the Differentially Expressed Genes between the Yellow and Black Stripes of the Juvenile Stripe in the Offspring of Wild Boar and Duroc

**DOI:** 10.3390/ani14142109

**Published:** 2024-07-19

**Authors:** Sanya Xiong, Dengshuai Cui, Naibiao Yu, Ruiqiu He, Haojie Zhu, Jiacheng Wei, Mingyang Wang, Wenxin Duan, Xiaoqing Huang, Liming Ge, Yuanmei Guo

**Affiliations:** National Key Laboratory for Swine Genetic Improvement and Germplasm Innovation, Jiangxi Agricultural University, Nanchang 330045, China

**Keywords:** wild boar, Duroc, juvenile stripes, coat color, whole-transcriptome sequencing

## Abstract

**Simple Summary:**

At birth, wild boars and their hybrid offspring usually have camouflage coat colors with juvenile stripes (the black/dark longitudinal stripes separated by yellow/light ones) on their back and two side flanks, and the juvenile stripes disappear at 4–5 months old. To study the mechanism of juvenile stripe maintenance, piglets with juvenile stripes were produced by crossing a wild boar with three Duroc sows. Through picture recording and pigment extraction, we observed that the stripe tended to disappear within approximately 70 days. Furthermore, whole-transcriptome comparison between two adjacent stripes (dark and light) on the back revealed that *ZIC4*, *ssc-miR-532-3p* and *ENSSSCG00000056225* might play a role in the formation of stripes. The present study laid the foundation for future research on the molecular mechanisms underlying juvenile stripes.

**Abstract:**

Coloration is a crucial trait that allows species to adapt and survive in different environments. Wild boars exhibit alternating black (dark) and yellow (light) longitudinal stripes on their back during their infancy (juvenile stripes), and as adults, they transform into uniform wild-type coat color. Aiming to record the procedure of juvenile stripes disappearing, piglets (WD) with juvenile stripes were produced by crossing a wild boar with Duroc sows, and photos of their coat color were taken from 20 d to 220 d. The pigments in the hairs from the black and yellow stripes were determined. Furthermore, the differentially expressed genes between the black and yellow stripes were investigated in 5 WD with the age of 30 d using whole-transcriptome sequencing to explore the genetic mechanism of the juvenile stripes. The juvenile stripes started to disappear at about 70 d, and stripes were not distinguished with the naked eye at about 160 d; that is, the juvenile stripe completely disappeared. A hotspot of a differentially expressing (DE) region was found on chromosome 13, containing/covering 2 of 13 DE genes and 8 of 10 DE lncRNAs in this region. A network among *ZIC4*, *ssc-miR-532-3p*, and *ENSSSCG00000056225* might regulate the formation of juvenile stripes. Altogether, this study provides new insights into spatiotemporal coat color pattern.

## 1. Introduction

Coat color is an important trait in pigs, and each breed has its own characteristic coat color. When coat color is segregated in a breed, the herdsman will consider the breed impure. To decode the genetic architecture of coat color, many studies have been carried out, and many causative genes have been identified, such as *MC1R* (melanocortin 1 receptor) [1], *KIT* (receptor tyrosine kinase) [2], and *TYRP1* (tyrosinase related protein 1) [3]. In most breeds, the coat color is constant throughout the whole life, but it is variable in wild boar and its intercrosses. At birth, the coat color of wild boars is a camouflage coat color with juvenile stripes, which becomes wild type (gray) as the wild boar grows. At 4–5 months old, the two kinds of stripes cannot be distinguished from each other with the naked eye [4]. The wild-type coat color is controlled by *MC1R* [5,6], but the causative gene controlling the camouflage coat color remains unknown.

To form a coat color pattern usually involves three steps, namely, establishment, implementation, and maintenance [7]. During the implementation and maintenance periods, melanocytes in hair follicles produce melanin particles (eumelanin and pheomelanin) and transmit them to the adjacent keratinocytes by melanosome [8]. This process is achieved through the autonomous regulation of melanocytes—*MC1R* [1], *SLC45A2* (solute carrier family 45 member 2) [9], and *CORIN* (serine peptidase) [10], and paracrine signals emanating from the dermis—*ASIP* (agouti signaling protein) and *EDN3* (endothelin 3) [11]). ASIP, a paracrine signaling molecule that inhibits the binding of *α*-MSH to the transmembrane MC1R protein of melanocytes, reduces the synthesis of eumelanin and increases the ratio of pheomelanin to eumelanin. *ASIP* has been reported to participate in the implementation of longitudinal stripes in avians [12], snowshoe hares [13], and dogs [14].

To decode the causative genes for coat color patterns, comparing the transcriptomes between dark and light stripe skins identified that *Alx3* (ALX homeobox 3) and *Sfrp2* (secreted frizzled-related protein 2) affected the striped formation of African stripe mice [15,16] and *DKK4* (dickkopf WNT signaling pathway inhibitor 4) was the causative gene for the coat color pattern of cats [17]. In addition to mRNA, miRNA and lncRNA play important roles in regulating the gene expression of coat color at the post-transcriptional level in mice [18], goats [19], alpacas [20], and pigs [21,22].

Samples are difficult to obtain due to wild boars being wild animals. In this study, we crossed a wild boar with Duroc sows to produce some piglets with juvenile stripes, and continuously observed the juvenile stripes from the age of 20 to 220 days to investigate its age of disappearance. To validate the observatory results, the melanin contents of the hairs from the dark and light stripes were recorded every 20 days. To decipher the causative genes of the juvenile stripe, we collected skin samples from adjacent areas of the dark and light stripes, and the whole-transcriptome sequencings of mRNA, miRNA, and lncRNA for each sample were determined. The mRNA, miRNAs, and lncRNA expressing differently between dark- and light-stripe skins were picked out, and the one involving pigment synthesis was the plausible candidate gene for juvenile stripes.

## 2. Materials and Methods

### 2.1. Animals

In this study, 19 piglets (WD) with juvenile stripes were produced by crossing a wild boar with 3 Duroc sows. In addition, a Shanxia long black pig [23] and a Duroc were used to collect the pure black and yellow hairs, respectively.

### 2.2. Photo and Sample Collection

#### 2.2.1. Photo Collection

To record the process of juvenile stripes disappearing, 2 of 19 F1 piglets, named WD01 and WD02, were photographed singly, and the rest from the 3 L were photographed with their littermates every 5 days from 20 to 160 d.

#### 2.2.2. Hair Collection

The hair samples from the dark and light stripes on the back of WD01 and WD02 were collected with scissors and stored in plastic bags every 10 days from 20 to 220 d. To give enough time for the hair growth, hairs were collected alternately on the left (L) and right (R) sides in three duplications (Figure 1). In addition, nine piglets were evenly divided into two groups, and hair samples were collected every 20 d from 40 to 120 d in one group and from 50 to 110 d in the other group. The pure black and pure yellow hairs were sampled from a Shanxia long black pig and a Duroc.

#### 2.2.3. Skin Tissue Collection

Five piglets were slaughtered at the age of 30 d, and the skin samples were collected from the dark and light stripes. A set of samples was dipped in RNAlater (Invitrogen) at 4 °C for two days and then stored at −80 °C for whole-transcriptome sequencing, and another set of samples was fixed with 4% paraformaldehyde for the histological study.

### 2.3. Phenotyping

#### 2.3.1. Counting the Hair Number of Each Color Category

The hair was divided into four categories, namely, pure black, pure yellow, wild type dominated by yellow, and wild type dominated by black (Figure 1), according to its color pattern [7]. The hair number of each category was counted for each hair sample.

#### 2.3.2. Quantifying the Pigment Content in Hairs

The hairs were washed with anhydrous ethanol to remove dirt and other foreign materials, and then put into 1.5 mL Eppendorf tubes. After drying overnight at 55 °C, the hairs were cut into powder with scissors. To mimic the different ratios of eumelanin to pheomelanin, black hair powder was mixed with yellow hair powder to produce mixed powder with 100, 90, 80, 70, 50, 30, 20, 10 and 0% black hair powder.

To determine the content of alkaline soluble melanin (ASM), 5 mg of hair powder were dissolved in 500 μL 1 mol/L NaOH, bathed in water at 85 °C for 4 h, and centrifuged at 11,000 rpm for 10 min to obtain supernatant. The optical density (OD) of the supernatant was measured using a spectrophotometer at a wavelength range of 250 to 700 nm, including 400 nm (Figure 1). In order to eliminate systematic error, 1 mol/L NaOH solution was used as a blank control. Each sample was measured three times, and the average of the three was considered its OD value.

To measure the content of eumelanin (EM) content, 5 mg of hair powder were put in a 1.5 mL Eppendorf tube, and 500 μL 30% hydriodic acid (HI) were added to hydrolyze pheomelanin (PM) in 80 °C water bath for 2 h. After restoring room temperature, 500 μL of 50% alcohol were added to the Eppendorf tube and centrifugated at 3000 rpm for 10 min, and the supernatant was discarded. The following steps were the same as the method to measure the content of EM described above (Figure 1).

#### 2.3.3. Hematoxylin–Eosin Staining

For each of the five piglets, 2 × 4 cm skin samples were dissected from the dark and light stripes on the back. The samples were dipped in 4% paraformaldehyde and fixed for 2 d. Then, they were embedded in paraffin and cut into 4 mm sections. The sections were stained with hematoxylin–eosin and their histological characteristics were examined under a light microscope. The skin thickness (excluding *stratum corneum*) was measured using a KFBIO Digital Slide Viewer (1.7.0.24). For each sample, the thickness was the average of skin thicknesses in 10 randomly selected visual fields.

### 2.4. Whole Transcriptome

#### 2.4.1. Extraction of RNA and Library Preparation

Total RNAs were extracted from each skin sample using TRIzol reagent (Invitrogen, Waltham, MA, USA) according to its manufacturer’s instructions. To analyze RNA quality, RNA concentration, 28S/18S, and RNA integrity number (RIN) were analyzed by an Agilent 2100 Bioanalyzer (Agilent, Santa Clara, CA, USA). The lncRNA (ribosomal depletion) and small RNA libraries were sequenced deeply by BGI on the DNBSEQ-PE100 and DNBSEQ-SE50 platforms, respectively.

#### 2.4.2. RT-qPCR Method

To remove genomic DNA, 3 μL RNA were reacted with 2.5 μL OligodT Primer in 7 μL DEPC water for 5 min at 70 °C. Then, 4 μL RT Buffer, 2 μL dNTPS, 1 μL RT Enzyme, and 0.5 μL RI Enzyme were added to the mixed liquid, and RNA was reversely transcribed into cDNA at 42 °C for 60 min and at 70 °C for 10 min. Finally, 0.2 μL cDNA were added to 9.8 μL RT-qPCR mix, which consisted of 5 μL SYBR Green I mix (2×), 0.2 μL Primer-F (10 μM), 0.2 μL Primer-R (10 μM), and 4.4 μL H_2_O, and amplified with the following program: 94 °C for 30 s; 40 cycles of 94 °C for 5 s, 60 °C for 15 s, and 72 °C for 10 s; 95 °C for 15 s, 60 °C for 60s, and 95 °C for 1 s. The forward (Primer-F) and reverse (Primer-R) primers of *ZIC4*, *ENSSSCG00000053388* and *ssc-miR-206* are shown in Appendix A.

### 2.5. Data Analysis

#### 2.5.1. Analyzing Pigment Content

A paired *t*-test was performed to test the difference of the pigments between the hairs from the black and yellow stripes using the *t.*test function, and the correlation coefficient between the OD value and the black hair proportion was calculated using the *cor.*test function in R (4.1.1).

#### 2.5.2. Analyzing LncRNA and mRNA

The analysis pipeline is shown in Figure 2. Raw data were scanned with SOAPnuke (v2.0) to remove the reads with the adapter sequences, as well as poly-N (>1%), and low-quality reads (Q20 > 40%). Subsequently, the reads were aligned to rRNA, and the matched reads were discarded to remove the rRNA sequences. The remaining reads were mapped to the porcine reference genome (*Sus scrofa* 11.1) using HISAT2 (v2.2.1) with the default parameters, and featureCounts (v2.0.3) and StringTie (v2.2.1) were used to summarize gene counts for each sample. The clear data were analyzed under a DESeq2 negative binomial generalized linear model, which was a powerful and robust approach to identifying differential expressing genes. We used FDR < 0.05 and |log2FoldChange| > 1 as threshold values to scan the differentially expressed genes (DEGs) with similar groups. Partial least squares discriminant analysis (PLS-DA) was conducted, and Venn diagram and heatmap were generated with R (R4.4.1).

#### 2.5.3. Analyzing microRNA

Raw data were also scanned with SOAPnuke to remove the reads with the adapter and poly-N (>1%), poly-A (>70%), and low-quality (Q13 > 10%) sequences. According to the size of the microRNA, only the reads from 18 to 29 bp were kept. Here, we used three mapping strategies. The clean reads were mapped to *Sus scrofa* 11.1 or the porcine mature miRNA database in miRBase (22.1) using Bowtie (1.3.1), and samtools (1.6) was used to analyze the genomic distribution and counts of microRNA. For comparison, the clean reads were mapped to the porcine reference genome using miRDeep2 (2.0.1.3), and microRNA was counted by quantifier.pl. The expression of microRNA was analyzed using DESeq2 (1.44.0), and the miRNA was classified as differentially expressed miRNA (DE-miRNA) if |log2FoldChange| > 1 and *FDR* < 0.05.

#### 2.5.4. Constructing lncRNA–miRNA–Gene Regulatory Networks

The prediction of DE-miRNA target genes was performed using miRanda (3.3a), TargetScan (7.0), and PITA, and lncRNA–miRNA–gene networks were inferred and visualized by Cytoscape (3.10.1).

## 3. Results

### 3.1. Changing of Juvenile Stripes from 20 to 160 d

Figure 3 showed the changing progress of juvenile stripes from 20 to 160 d. The light and dark stripes were clearly distinguished from each other with the naked eye before 110 d, and then the difference between them became blurry and completely disappeared at 160 d. The light stripes did not become the dark stripes, and vice versa; both of them became the wild-type coat color (dark brown).

The hair was classified into pure black, pure yellow, yellow-black, and black-yellow, and their proportions were calculated based on their number in a 2 × 2 cm^2^ area during pig growth. Figure 4A illustrates the dynamic changes in the proportions with the growth of WD01 and WD02. The proportion of black hairs in the dark stripes slightly increased at first and then remained invariant during 20–220 d. In the light stripes, the proportion of black hairs showed a rapid increase at approximately 70 d and then remained unchanging. Figure 4B shows the dynamic changes in the proportions with the growth of another nine pigs from 40 to 120 d. The difference between the light and dark stripes increased at first and then gradually decreased at 70 d, and this changing tendency was similar to that observed in Figure 4A. Both in the light and dark stripes, the proportion of pure black and pure yellow decreased (Figure 4C), while the proportion of yellow-black and black-yellow gradually increased.

### 3.2. Pigment Contents in the Hairs of Light and Dark Stripes

The extracted alkaline-soluble melanin (ASM) was a complex, with light absorption peaks in the range of 350 to 450 nm (Appendix A). In addition, OD_400_ was strongly correlated with the proportion of black hair power (*r*_400_ = 0.868, *p* = 1.238 × 10^−10^), indicating that OD_400_ could be used to measure the content of ASM (Appendix A), which was similar to taking OD_400_ as the characteristic absorption peak of ASM in alpaca hair fiber [24]. The OD_350_ value can be used to determine the content of eumelanin (*r*_350_ = 0.918, *p* = 4.981 × 10^−14^, Appendix A). Solvene-350 reagent, another alkaline reagent, was used to dissolve the sheep and human hair, and the OD_500_ and OD_650_ values were used to measure the contents of ASM and eumelanin, respectively [25]. The OD_500_ and OD_650_ values significantly correlated with the proportion of black hair power (*r*_500_ = 0.936, *p* = 3.913 × 10^−15^; *r*_650_ = 0.893, *p* = 2.979 × 10^−12^), but their sensitivities were weaker than that of 350 nm here (Appendix A).

The hair ASM and EM contents of the dark and light stripes were determined using the OD_400_ and OD_350_ values, respectively. In both light and dark stripes, the ASM contents increased at first and then decreased (Figure 5A), while the EM contents decreased at first and then increased (Figure 5B).

### 3.3. Histological Differences between the Light and Dark Stripes

The skin thicknesses and hair follicle numbers were compared between the light and dark stripes at about one month old. The average dermal thickness was 1590 μm at the dark stripes and significant thicker (*p* = 0.005, Figure 6A) than 1347 μm at the light stripes. In the juvenile stripe area, three hair follicles were clustered into a hair follicle unit, which is profiled with dashed lines in Figure 6B,C. Eumelanin was clearly visible in the black hair follicle (Figure 6D), but it was masked by pheomelanin in the yellow hair follicles (Figure 6E). Both the thickness of the epidermis without *stratum corneum* (*p* = 0.734, Figure 6H) and the hair follicle density (*p* = 0.571, Figure 6I) showed no significant difference between the light and dark stripes, although the latter had large values. The color difference between the light and dark stripes might be due to differences in pigment types and their amounts.

### 3.4. Differentially Expressing mRNA and lncRNA between the Light and Dark Stripes

For each sample, at least 85.3 million reads passed the quality control (Table 1), and the Q30 of those qualified reads (clear reads) was greater than 95%. The average GC content of the clean reads was between 54 and 57%, and the mapping rates of the 10 samples were all greater than 80%.

To decode the maintaining mechanism of juvenile stripes, the transcriptome differences between light and dark stripes were investigated through whole-transcriptome sequencing. The expression patterns of mRNA and lncRNA showed more differences among individuals than between stripes (Appendix A). Using featureCounts soft, 13 genes and 10 lncRNAs (*ENSSSCG00000053718*, *ENSSSCG00000058666*, *ENSSSCG00000052873*, *ENSSSCG00000053388*, *ENSSSCG00000061303*, *ENSSSCG00000055695*, *ENSSSCG00000058850*, *ENSSSCG00000056225*, *ENSSSCG00000063027*, *ENSSSCG00000056963*) showed differential expression between the light and dark stripes (Figure 7A,B), and the PC1 explained 76% of the variation between the stripes (Appendix A).

The logarithms of fold changes with a base of 2 (log2FC) of *ZIC4* (Zic family member 4), *PAX9* (paired box 9), *ESR1* (estrogen receptor 1), *PECR* (peroxisomal trans-2-enoyl-CoA reductase), *MYH11* (myosin heavy chain 11), *RYR3* (ryanodine receptor 3), *ZIC1* (Zic family member 1), *ENSSSCG00000043038*, *CKM* (creatine kinase, M-type), *ZIC2* (Zic family member 1), *VSIG1* (V-set and immunoglobulin domain-containing 1), and *KCNE5* (potassium voltage-gated channel subfamily E regulatory subunit 5) were significantly downregulated (log2FC ≤ −1, FDR < 0.05), while *KRT84* (keratin 84) was significantly upregulated in the light stripes (log2FC = 1.51, FDR = 6.73 × 10^−8^). Except for *VSIG1* and *ZIC2*, all of the DEGs were also detected by StringTie soft (Appendix A).

Eight of the ten DE-lncRNAs were significantly downregulated in the light stripes, and all of them were located near two DEGs (*ZIC4* and *ZIC1*) on chromosome 13 (Figure 7B,C). The qPCR results validated that the downregulating expression of *ENSSSCG00000053388* and *ZIC4* took place in the light stripes rather than in the dark stripes (Figure 7D). Although the fragments per kilobase million (FPKM) of coat color genes were not significantly different between the light and dark stripes, the FPKM of *TYR*, *TYPR1*, and *DCT* was higher in the light stripes than in the dark stripes, and vice versa for the FPKM of *ASIP* (Figure 7E).

### 3.5. Differentially Expressed miRNAs and Their Predicted Target Genes

The sizes of clean reads used to detect the DE-miRNA were from 18 to 29 bp, concentrated in 22–23 bp (Figure 8A). With two mismatches, more than 83% of clean reads were mapped to the reference genome (Table 2). Due to the small size of miRNAs (about 18–24 bp), mismatches were not permitted. Except for two samples, the mapping rates of all samples were more than 75%, with an average mapping rate of 79.74%. Although some reads were mapped to other types of RNAs, 97.87% of reads were mapped to miRNAs (Appendix A). There were 457 mature and 536 hairpin miRNAs of pigs in the miRBase database, and more than 44% clean reads were matched to the miRNAs on the miRBase database.

Two programs (samtools and mirdeep2) and two datasets (all samples with and without WD12_L and WD13_D) were used to detect the DE-miRNAs. Regardless of the program and dataset, four miRNAs were found to be differentially expressed in the light and dark stripes. In the light stripes, ssc-miR-335, ssc-miR-532-3p, and ssc-miR-551a were significantly upregulated, and ssc-miR-206 was significantly downregulated. Except for the four DE-miRNAs mentioned above, ssc-miR-148 and ssc-miR-1 were found to be significantly upregulated and downregulated, respectively, in the light stripes using samtools in all samples (Figure 8B,C).

The relationship between the DE-mRNAs and DE-miRNAs was predicted using miRanda, TargetScan, and PITA, and five common significant mRNA-miRNA pairs, namely, ssc-miR-195—*RYR3*, ssc-miR-1—*ZIC4*, ssc-miR-206—*ZIC4*, ssc-miR-335—*ESR1*, and ssc-miR-532-3p—*ZIC4* (Figure 8D and Figure 9), were identified. In mammals, miRNAs typically bind to the 3′UTR of the target mRNA and inhibit its translation by reducing its stability. Based on the predicted results and the differential gene expression between the light and dark stripes, *ZIC4* and ssc-miR-532-3p may be involved in the formation of juvenile stripes.

## 4. Discussion

Wild boars are mainly found in shrubs, and the juvenile stripe serves as camouflage when the piglets are young and cannot protect themselves. Crossing Duroc sows with wild boars produces piglets with the juvenile stripe. In this study, 19 piglets from a cross between wild boar and Duroc were used to speculate the disappearing age of juvenile stripes and the differentially expressed mRNAs, lncRNAs, and miRNAs between the light and dark stripes. The juvenile stripes disappeared gradually, and the light and dark stripes could not be distinguished from each other with the naked eye at about 5–6 months, and 13 mRNAs, 10 lncRNAs, and 4 miRNAs were observed to be differentially expressed between the light and dark stripes at about 1 month.

The coat color in mammals is usually recorded qualitatively, such as describing the irregular color of the whole body [26]. However, it can be recorded semi-quantitatively by counting the number of each hair type, or quantitatively by measuring the contents of two types of hair pigments. As early as 1995, high-performance liquid chromatography (HPLC) and spectrophotometry were used to determine the hair melanin in 10 mutant mice with various coat color [27], and then this method was used to measure the hair melanin in alpacas [24] and tigers [10], and in cuttlefish [28]. But until now, it has not been reported in pigs. In this study, taking photos, counting the hair number, and determining the pigment content were used to record the coat color. All of them gave similar results that the juvenile stripes started to disappear at about 70 d and disappeared completely at about 5–6 months, which is consistent with the result of the juvenile stripes disappearing within 2–5 months in wild boars [29].

In this study, the OD values of pigment extracts were recorded with wavelengths between 250 and 700 nm, and the OD values of 350, 400, 500, and 650 nm strongly correlated (*p* ≤ 1.238 × 10^−10^) with the proportion of black hair (*r*_350_ = 0.918, *r*_400_ = 0.868, *r*_500_ = 0.936, and *r*_650_ = 0.893). These results validated the reliability of using 350 and 400 nm (but not limited to these) to measure the pigment content. Furthermore, the combination of 350 and 400 nm ultraviolet light is commonly used when using the NaOH alkaline solution method [24,27]. Therefore, the 350 and 400 nm ultraviolet lights were chosen to determine the EM and ASM content, respectively, in this study. There were also some other references using 500 and 650 nm visible light to measure the content of total melanin (TM), for example, TM in raccoon dog skin [30], but with other alkaline reagents.

The difference in pigment contents between the light and dark stripes began to decrease at the age of 70 d (Figure 5A), similar to the proportion of the black hairs (Figure 4A,B). The hair follicle is a complex micro-organ that produces melanin and undergoes a three-period cyclic process, namely, anagen, catagen, and telogen. About 95% of the hair follicles were in anagen at birth and then decreased to about 50% after seven weeks. Afterwards, anagen hair follicles showed an upward trend with piglet growth [31], and prenatal hair follicle morphogenesis development was maintained until the 45th day after birth (45 d) [32]. To compare transcriptome differences between dark and light stripes more accurately, the skin of striped pigs was selected for subsequent studies before 45 d. This choice was also partially supported by the histological experiment in this study. At about 30 d, the average dermal thickness was 1590 μm in the dark stripe and significantly thicker (*p* = 0.005, Figure 6A) than 1347 μm in the light stripe, similar to the results observed in striped cats [15]. Furthermore, eumelanin was clearly visible in the former (Figure 6E), but it was masked by pheomelanin in the latter (Figure 6F).

The RNA integrity numbers (RINs) of samples were between 5.1 and 7.6, indicating partial degradations of RNAs. Previous studies indicated that the whole transcriptome expression profile of the sample with RIN = 7.2 was similar to the sample with RIN = 2.5 from the same individual, and RNA-degraded samples with RIN values > 4.4 could still be used to perform gene expression analysis [33,34,35]. Therefore, the expression profiles obtained from the samples with RINs between 5.1 and 7.6 were still reliable.

Among the 13 differentially expressed genes, Genes from the *ESR1* and ZIC families (include *ZIC1* and *ZIC4*) are related to coat color. *ESR1* is expressed in the melanocytes and plays an important role in pigment synthesis [36,37]. *ZIC4*, as a selector gene, is related to the dorsal–ventral pattern in the teleost trunk, such as pigmentation patterns [38,39]. In the African striped mouse, the *ZIC4* gene exhibits no differential expression among all dorsal dark and light stripes, indicating that it is not a major contributor to dorsal stripe formation but rather plays a role in overall dorsal pattern development [15]. Furthermore, 8 of the 10 DE-lncRNAs are close to *ZIC4* to within 300 kb (Figure 7C). Among the seven DE-miRNAs, *ssc-miR-532-3p* probably participates in the formation of the dorsal pattern in striped pigs through interacting with *ZIC4* and *ENSSSCG00000056225* (Figure 9).

There are few genes expressed differentially between the light and dark stripes but no well-known causative gene of coat color. For example, *MC1R*, *TYR*, *TYRP1*, *DCT*, and *ASIP*, show a significant signal of differential expression. One reason is that they may be not the causative genes for the juvenile stripes, and another reason is that those genes are not expressed significantly differently between the light and dark stripes at 30 d. The juvenile stripe is spatiotemporal, and it disappears as the piglet grows. Although we have infered that the juvenile stripe is maintained at least for 30 d, the difference in gene expression between the light and dark stripes may not last for so long, because there is a lag difference between the pigment appearing in the hairs and the gene expression of coat color. Furthemore, the juvenile stripe disappears gradually, and the difference in the causative gene expression between the light and dark stripes also gradually decreases. At 30 d, the difference may not reach the significant level.

*ASIP*, which regulates pigment synthesis directly, is involved in the formation of stripes in zebrafish and quail by competitively combining *MC1R* [40,41]. Japanese quails have black and yellow stripes, and *ASIP* is expressed in the yellow stripe area during embryonic development. Although there was no significantly differentially expression between the light and dark stripes, *ASIP* was expressed more in the dark stripes and *MC1R* was expressed more in the light stripes (Figure 7E), and this indicates that earlier than the results of the phenotypic observation, there was already a trend of the disappearance of stripes at 30 d. It is essential to continue determining the disappearance time of juvenile stripe. Additionally, we will conduct a comprehensive analysis, including the validation of *ZIC4* function and the regulatory network, as well as epigenetic regulatory pathways.

## 5. Conclusions

The juvenile stripe showed a trend of disappearance at about 70 d, and it completely disappeared at about 160 d by phenotypic recording. A hotspot of the differentially expressed region between the light and dark stripes was found on chromosome 13, containing/covering two units of 13. A network among *ZIC4* (DE-gene), *ssc-miR-532-3p* (DE-miRNA), and *ENSSSCG00000056225* (DE-lncRNA) might regulate the juvenile stripe.

## Figures and Tables

**Figure 1 animals-14-02109-f001:**
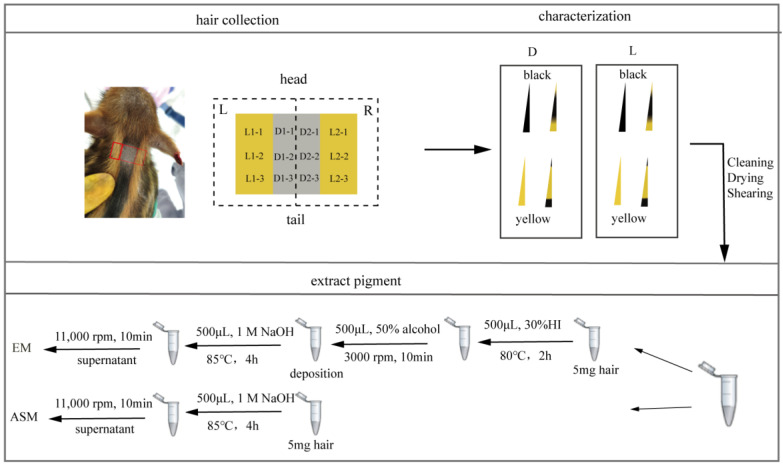
Schematic of hair collection and processing. The red box corresponds to where the hair was collected.

**Figure 2 animals-14-02109-f002:**
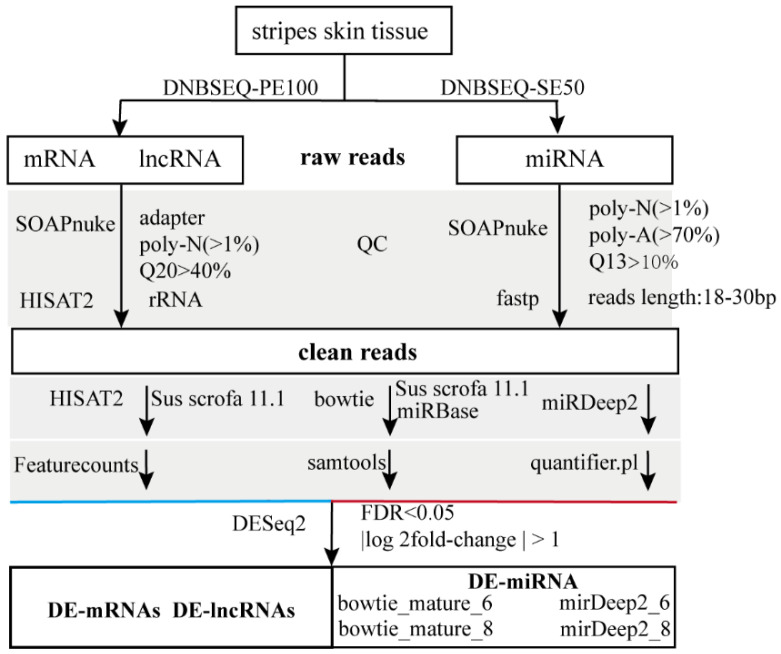
Transcription analysis pipeline.

**Figure 3 animals-14-02109-f003:**
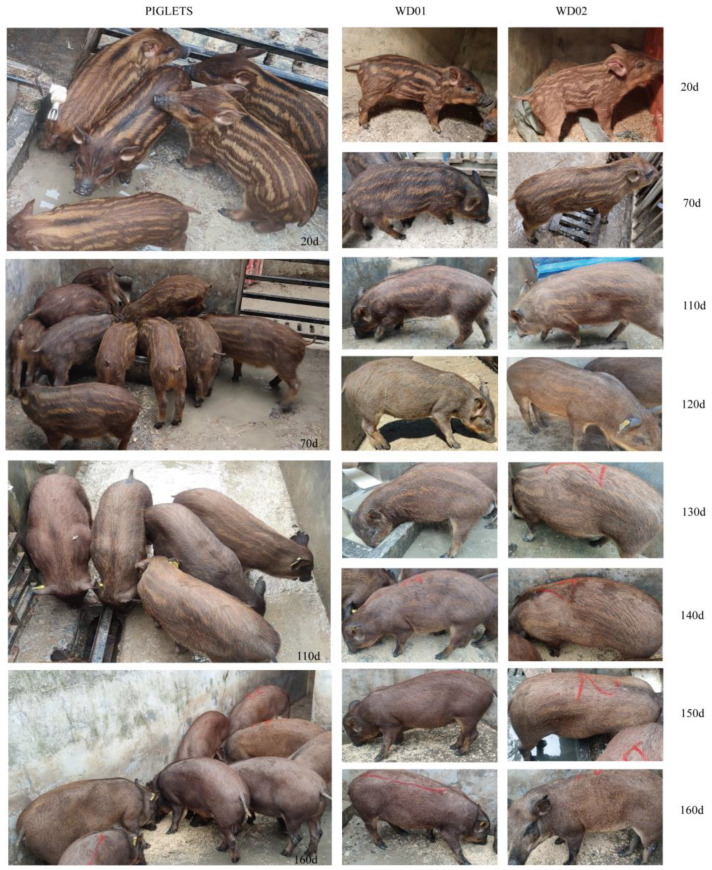
Changing of juvenile stripes from 20 to 160 days. WD01 and WD02 were littermates.

**Figure 4 animals-14-02109-f004:**
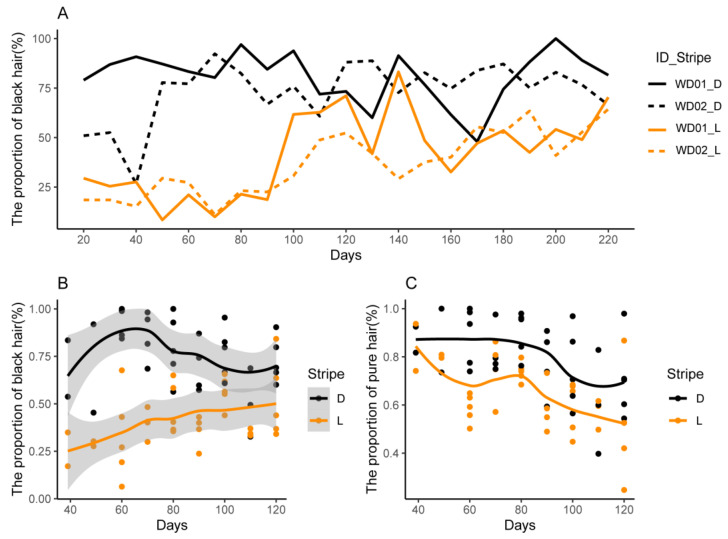
Dynamic change of hair types in the light and dark stripes. (**A**) The proportion of black hair of WD01 and WD02; (**B**) the proportion of black hair in 9 other pigs; (**C**) the proportion of pure black and pure yellow hairs in 9 other pigs.

**Figure 5 animals-14-02109-f005:**
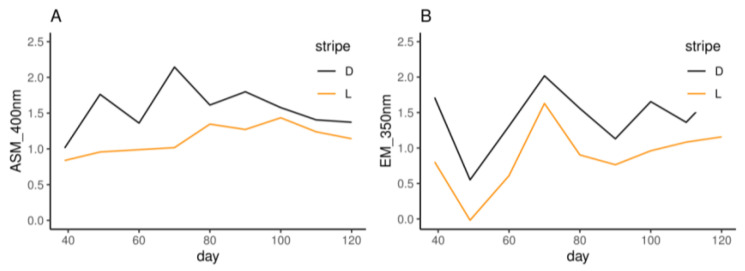
The absorbance values of alkaline-soluble melanin (ASM, **A**) at 400 nm and eumelanin (EM, **B**) at 350 nm. L and D indicate the light and dark stripes, respectively.

**Figure 6 animals-14-02109-f006:**
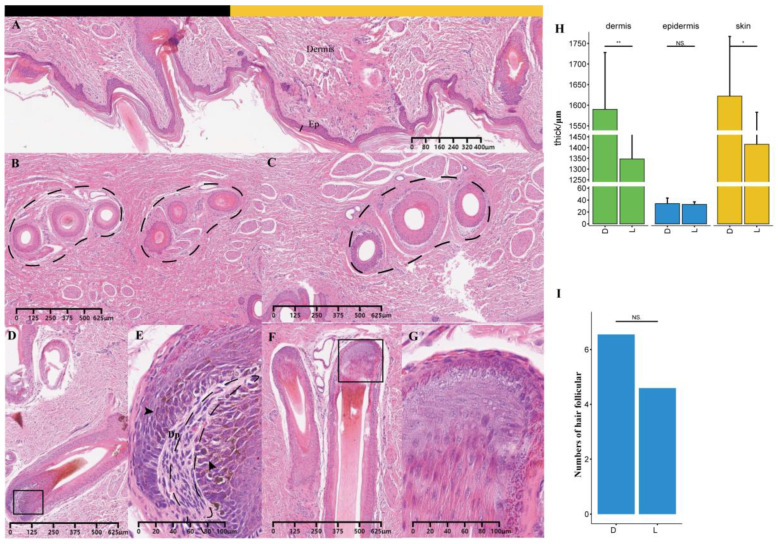
The histological characteristics of skin and hair follicles in the juvenile stripe area. (**A**) Skin with dark (black block) and light (yellow block) stripes (4 × 10). Ep: epidermis. (**B**,**C**) Cross-sections of hair follicles in the dark and light stripes, respectively (4 × 10). (**D**,**F**) Hair follicles (4 × 10) in the dark and light stripes, respectively. (**E**,**G**) Hair follicles (20 × 10) in the dark and light stripes, respectively. (**H**,**I**) Differences in skin thickness and number of hair follicles per centimeter of skin section between dark and light stripes, respectively. The black square indicates the position of local amplification; * *p* ≤ 0.05; ** *p* ≤ 0.01; NS, *p* > 0.05.

**Figure 7 animals-14-02109-f007:**
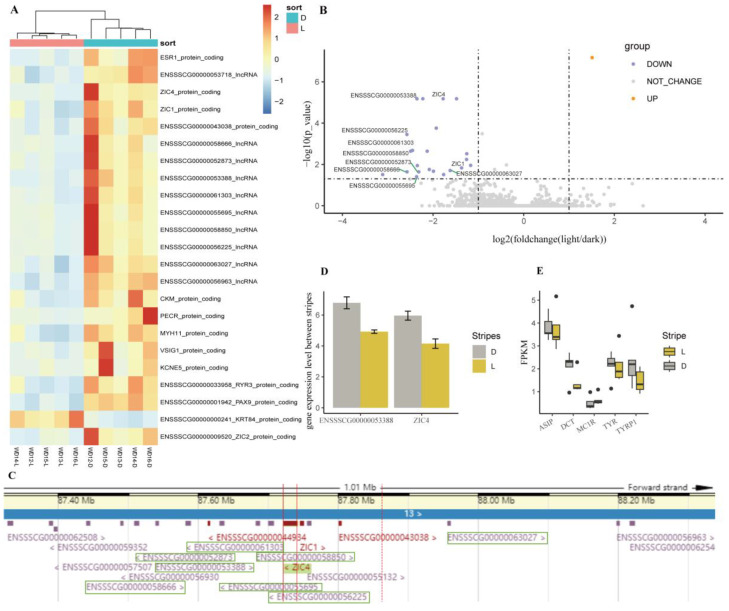
The expression of mRNA and lncRNA between the light and dark stripes. (**A**) Heatmap. (**B**) Volcano plot. Purple indicates genes that were downregulated in light stripes, while orange indicates genes that were upregulated in light stripes. The names of DE-lncRNAs near *ZIC4* and *ZIC1* are given. (**C**) The genes and lncRNAs near *ZIC4* (chr13:87722146-87740921). The DE-lncRNAs are indicated with a green box. (**D**) Quantifying the expression of *ZIC4* and ENSSSCG00000053388 by RT-qPCR. (**E**) The FPKM of *ASIP*, *DCT*, *MC1R*, *TYR*, and *TYRP1*.

**Figure 8 animals-14-02109-f008:**
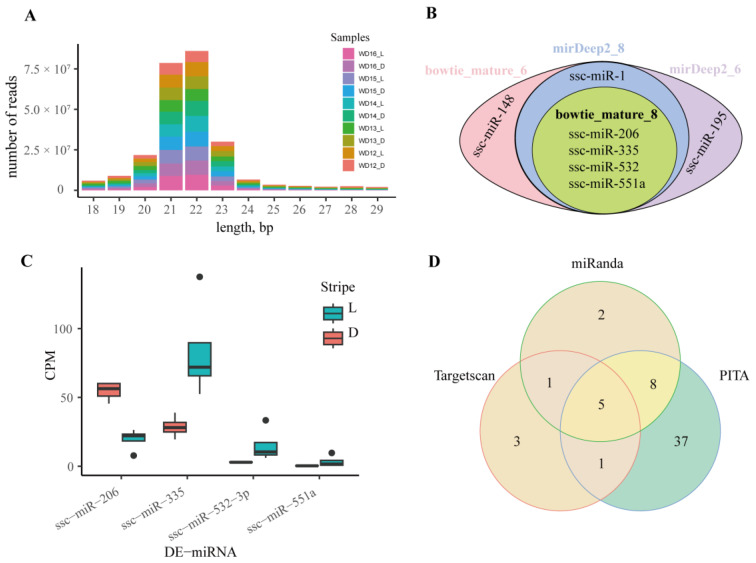
Differentially expressed miRNA between the light and dark stripes. (**A**) The histogram of clean reads. (**B**) The Venn diagram of DE-miRNAs. Bowtie_mature_8 and Bowtie_mature_6: the DE-miRNAs identified by mapping clean data of all samples without WD12_L and WD13_D and of samples for 32d to the reference miRNAs using Bowtie and samtools, respectively; mirDeep2_8 and mirDeep2_6: the DE-miRNAs identified by mapping clean data of all samples with and without WD12_L and WD13_D to reference miRNAs using mirDeep2, respectively. (**C**) Counts of exon model per million mapped reads (CPM) of DE-miRNAs in bowtie_mature_8. (**D**) Venn diagram of the miRNA target genes predicted by Targetscan, miRanda, and PITA.

**Figure 9 animals-14-02109-f009:**
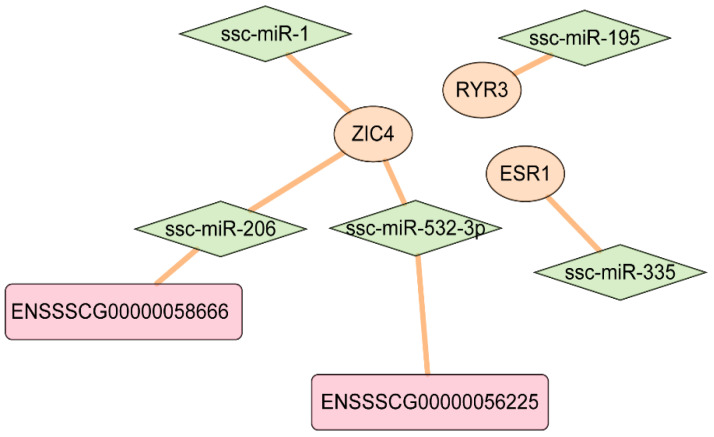
The predicted interaction network among lncRNAs, miRNAs, and genes.

**Table 1 animals-14-02109-t001:** The sequencing quality and mapping rate of mRNA and lncRNA.

Sample	Clean Reads	Error Rate, %	Q30, %	GC, %	Mapped Reads, %	Uniquely Mapped Reads
WD12_D	86,785,883	20	95.38	54.0	93.78	72,522,887 (83.57%)
WD12_L	86,030,558	20	95.04	54.0	83.06	61,768,321 (71.80%)
WD13_D	85,300,011	20	95.48	54.0	89.65	67,497,154 (79.13%)
WD13_L	85,408,475	20	95.41	55.0	93.56	69,157,672 (80.97%)
WD14_D	84,933,279	20	95.35	55.0	96.85	70,709,996 (83.25%)
WD14_L	85,725,420	30	95.10	57.0	96.23	68,646,288 (80.08%)
WD15_D	86,755,681	20	95.22	56.0	96.61	72,592,898 (83.68%)
WD15_L	85,916,341	20	95.52	55.0	96.10	71,438,074 (83.15%)
WD16_D	85,440,535	20	95.36	55.0	95.52	69,992,205 (81.92%)
WD16_L	85,452,301	20	95.6	55.0	95.95	71,610,674 (83.80%)

**Table 2 animals-14-02109-t002:** The sequencing quality and mapping rate of miRNA.

Sample	Raw Counts	Clean Reads	Average Read Length, bp	Q30, %	Mapped Reads within Two Mismatches, %	Mapped Reads without Mismatch, %	
						Genome	Mature
WD12_D	27,200,000	25,301,429	21.83	96.97	95.03%	75.95%	54.28%
WD12_L	27,680,000	25,256,111	22.17	97.48	83.72%	63.05%	44.63%
WD13_D	29,600,000	25,184,809	21.89	96.89	86.67%	64.23%	45.63%
WD13_L	27,200,000	25,272,976	21.81	97.58	94.90%	76.68%	55.61%
WD14_D	27,360,000	25,378,446	21.62	97.15	98.44%	79.69%	60.59%
WD14_L	27,680,000	25,151,135	21.74	97.46	97.74%	78.90%	50.34%
WD15_D	27,520,000	25,173,779	21.74	97.04	98.38%	80.69%	56.40%
WD15_L	27,840,000	25,169,469	21.58	97.16	98.59%	81.79%	58.35%
WD16_D	27,200,000	24,988,396	21.90	96.75	98.18%	80.89%	55.75%
WD16_L	28,480,000	25,193,907	21.75	96.97	97.89%	81.13%	55.00%

## Data Availability

The data presented in this study are available upon request from the corresponding author.

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
