# Peer review of "Exploring the Maintaining Period and the Differentially Expressed Genes between the Yellow and Black Stripes of the Juvenile Stripe in the Offspring of Wild Boar and Duroc"

_animals, 2024, doi:10.3390/ani14142109_

Round 1

Reviewer 1 Report

Comments and Suggestions for Authors

Submitted paper entitled "Exploring the maintaining period and the differentially expressed genes between the yellow and black stripes of the juvenile stripe in the offspring of wild boar and Duroc" in my opinion is very interesting, good written, analytical methods are well chosen, similar to bioinformatics tools.

As a reviewer, however I have some comments, which will improve article making its more clear.

·         Line 13 (boars) vs. 76 (boar) - how many boars were crossed with 3 Duroc sows?

·         Line 30 - Add underlined word and shortcut in the sentence "A hotspot of differentially expressing (DE) region was found on chromosome 13, containing/covering two of 13 DE-genes and 8 of 10 DE-lncRNAs in this region."

·         Line 32 - italicize all genes names

·         Lines 40, 51-52, 59, 265-267 - explain gene names

·         Line 49 - replace "product" by "produce"

·         Line 59 - Did all 3 genes - Alx3, Sfrp2 and DKK4 were causative genes for coat color pattern in both species?

·         Line 63 - replace "to Duroc" by "with Duroc"

·         Line 110-122 - proper English tense must be applied in the whole part! e.g. was measured, was added, was used...

·         Lines 128, 237 - italicize latin name - stratum corneum

·         Line 136 - replace "LncRNA" by "lncRNA"

·         Line 144 - replace "was consist" by "consisted"

·         Line 145 - H2O - 2 in subscript

·         Line 157 - replace "was" by "is"

·         Figure 3 - did authors photographed only SOWS (word above left photos)? In the text there is no info about sex of piglets. I think it should be changed on PIGLETS

·         Line 256 - replace "different" by "differences"

·         Line 263 - move sentence "Except for VSIG1 and ZIC2, all of the DEGs were also detected by StringTie soft (Table S2) after sentence in lines 265-268

·         Line 270 - replace "them located" by " them were located"

·         Line 326 - replace "to wild" by "with wild"

·         Line 359 - I am not sure if part of sentence "active or anagen at birth" is correct; maybe "active in anagen at birth"

·         Lines 361-363 - explain more precisely terms "Postnatal P45" and "before 45d"

·         Line 382 - replace "In" by "Among"

·         Line 382 - italicize gene symbol

·         Line 382 - replace "maybe" by "probably"

·         Line 410 - add underlined word "as found on chromosome 13, containing/covering two of 13"

·         Lines 411-412 - italicize all genes symbols

·         Table S2 - replace "Differently expression" by "Differently expressed"

CONGRATULATIONS, IT IS REALLY INTERESTING PAPER, I HAVE READ IT WITH GREAT PLEASURE!!! GOOD LUCK WITH ITS PUBLICATION!!!

Comments on the Quality of English Language

Section 2.3.2 must be corrected to give proper tense

Author Response

For research article

Response to Reviewer 1 Comments

1. Summary

Thank you for your valuable feedback on our manuscript. We appreciate the time and effort you have put into carefully reviewing our work. Below, we have addressed each of your comments and provided the corresponding revisions in the re-submitted files.

2. Questions for General Evaluation

Reviewer’s Evaluation

Response and Revisions

Does the introduction provide sufficient background and include all relevant references?

Yes/Can be improved/Must be improved/Not applicable

Is the research design appropriate?

Yes/Can be improved/Must be improved/Not applicable

Are the methods adequately described?

Yes/Can be improved/Must be improved/Not applicable

Are the results clearly presented?

Yes/Can be improved/Must be improved/Not applicable

Are the conclusions supported by the results?

Yes/Can be improved/Must be improved/Not applicable

3. Point-by-point response to Comments and Suggestions for Authors

Comments 1: Submitted paper entitled "Exploring the maintaining period and the differentially expressed genes between the yellow and black stripes of the juvenile stripe in the offspring of wild boar and Duroc" in my opinion is very interesting, good written, analytical methods are well chosen, similar to bioinformatics tools. As a reviewer, however I have some comments, which will improve article making its more clear.

Response 1: Thank you for your encouragement of our research and manuscript. Meanwhile, thank you for your detailed revision suggestions.

Comments 2: Line 13 (boars) vs. 76 (boar) - how many boars were crossed with 3 Duroc sows?

Response 2: Thanks for your question. After re-checking the information, it was determined that a boar was successfully crossed with 3 Duroc sows. We have revised it in the manuscript and highlighted it in red.

ž   Line 13 (Line 14): “crossing a wild boar with three Duroc sows”

ž   Line 66: “we crossed a wild boar with Duroc sows”

Comments 3: Line 30 - Add underlined word and shortcut in the sentence "A hotspot of differentially expressing (DE) region was found on chromosome 13, containing/covering two of 13 DE-genes and 8 of 10 DE-lncRNAs in this region."

Response 3: I'm unsure if we are correctly understanding your intention. We created a link for this sentence: When clicked, the page redirects to Figure 5.

ž   Lines 30-31: “A hotspot of differentially expressing (DE) region was found on chromosome 13, contain-ing/covering two of 13 DE-genes and 8 of 10 DE-lncRNAs in this region”

Comments 4: Line 32 - italicize all genes names.

Response 4: We have revised the genes in “A network among ZIC4, ssc-miR-532-3p and ENSSSCG00000056225” to italics. You can also see this in the resubmitted document: line 32

Comments 5: Lines 40, 51-52, 59, 265-267 - explain gene names.

Response 5: Agree. We have added the full gene names in resubmitted manuscript. The contents in parentheses represent the corresponding lines modified in the resubmitted manuscript.

ž   Line 40 (Lines 40-41 ):”MC1R (melanocortin 1 receptor), KIT (receptor tyrosine kinase), and TYRP1 (tyrosinase related protein 1)”

ž   Lines 51-52 (Lines 52-54):“This process is achieved through the autonomous regulation of melanocytes—MC1R, SLC45A2 (solute carrier family 45 member 2) and CORIN (serine peptidase), and paracrine signals emanating from the dermis—ASIP (agouti signaling protein) and EDN3 (endothelin 3)”

ž   Line 59 (Lines 60-62 ):“Alx3 (ALX homeobox 3), Sfrp2 (secreted frizzled related protein 2) and DKK4 (dickkopf WNT signaling pathway inhibitor 4) were causative genes for coat color pattern”

ž   Lines 265-267 (Lines 266-273 ): “The logarithms of fold changes with base of 2 (log2FC) of ZIC4 (Zic family member 4), PAX9 (paired box 9), ESR1 (estrogen receptor 1), PECR (peroxisomal trans-2-enoyl-CoA reductase), MYH11(myosin heavy chain 11), RYR3 (ryanodine receptor 3), ZIC1 (Zic family member 1), ENSSSCG00000043038, CKM(creatine kinase, M-type), ZIC2(Zic family member 1), VSIG1(V-set and immunoglobulin domain containing 1) and KCNE5(potassium voltage-gated channel subfamily E regulatory subunit 5) were significantly down-regulating (log2FC ≤ -1, FDR < 0.05), while KRT84 (keratin 84) was significantly up-regulating in the light stripe(log2FC=1.51, FDR=6.73E-08).

Comments 6: Line 49 - replace "product" by "produce"

Response 6: Agree. You can see that in the re-submitted manuscript.

ž   Line 49 (Line 50): ”melanocytes in hair follicles produce melanin particles”

Comments 7: Line 59 - Did all 3 genes - Alx3, Sfrp2 and DKK4 were causative genes for coat color pattern in both species?

Response 7: The three genes were not causative genes for coat pattern in both species. In striped mice, it was identified that Alx3 and Sfrp2 affected the formation of striped pattern. In cats, DKK4 was identified to participate in the formation of coat color pattern. Because of the ambiguity caused by our incorrect expression, we have modified it in the re-submitted manuscript.

ž   Line 59 (Lines 59-63): “To decode the causative genes for coat color patterns, comparing the transcriptomes between dark and light stripe skins had identified that Alx3 (ALX homeobox 3), Sfrp2 (secreted frizzled related protein 2) affected the striped formation of African stripe mice[15,16]  and DKK4 (dickkopf WNT signaling pathway inhibitor 4) was causative gene for coat color pattern of cat“

Comments 8: Line 63 - replace "to Duroc" by "with Duroc"

Response 8: Agree. You can see that in the re-submitted manuscript.

ž   Line 63 (Line 66): “we crossed a wild boar with Duroc sows”

Comments 9: Line 110-122 - proper English tense must be applied in the whole part! e.g. was measured, was added, was used...

Response 9: Thank you for pointing out our English tenses. We have corrected it in the resubmitted manuscript.

ž   Line 110-122 (Lines 113-125): “To determine the content of alkaline soluble melanin (ASM), 5mg of hair powder was dissolved into 500 μL 1 mol/L NaOH, bathed in water at 85 ℃ for 4 hours, and centrifuged at 11,000 rpm for 10 minutes to obtain supernatant. The optical density (OD) of the supernatant was measured using a spectrophotometer at a wavelength range of 250 to 700 nm, including 400 nm (Figure 1).  In order to eliminate systematic error, 1 mol /L NaOH solution was used as a blank control. Each sample was measured three times, and the average of the three was considered its OD value.

ž   To measure the content of eumelanin (EM) content, 5mg of hair powder was put into 1.5mL Eppendorf tube, and 500 μL 30% hydriodic acid (HI) was added to hydrolyze pheomelanin (PM) in 80 ℃ water bath for 2h. After restoring room temperature, 500 μL of 50% alcohol was added to the Eppendorf tube, centrifugated at 3000 rpm for 10 minutes, and the supernatant was discarded. The following steps were the same as the method to measure the content of EM described above (Figure 1).”

Comments 10: Lines 128, 237 - italicize latin name - stratum corneum.

Response 10: Agree. You can see that in the re-submitted manuscript.

ž   Line 128 (Line 131): “The skin thickness (excluding stratum corneum)”

ž   Line 237 (Line 240): “Both the thickness of epidermal without stratum corneum

Comments 11: Line 136 - replace "LncRNA" by "lncRNA".

Response 11: Agree. You can see that in the re-submitted manuscript.

ž   Line 136 (Line 139): “The lncRNA (ribosomal-depletion)”

Comments 12: Line 144 - replace "was consist" by "consisted".

Response 12: Agree. You can see that in the re-submitted manuscript.

ž   Line 144 (Line147): “which consisted of 5 μL SYBR Green I mix (2×)”

Comments 13: Line 145 - H2O - 2 in subscript.

Response 13: Agree. You can see that in the re-submitted manuscript.

ž   Line 145 (Line 148): “4.4 μL H2O”

Comments 14: Line 157 - replace "was" by "is".

Response 14: Agree. You can see that in the re-submitted manuscript.

ž   Line 157 (Line 160): “The analysis pipeline is showed in figure 2”

Comments 15: Figure 3 - did authors photographed only SOWS (word above left photos)? In the text there is no info about sex of piglets. I think it should be changed on PIGLETS.

Response 15: The pig herd shown on the left is of both sexes and includes sows and boars. Like you said, it doesn't focus on gender either. Because we were careless before, it affected the meaning of the figure 3. In the resubmission, we have changed "SOWS" to "PIGLETS”. Line 196:

Comments 16: Line 256 - replace "different" by "differences".

Response 16: Agree. You can see that in the re-submitted manuscript.

ž   Line 256 (Line 258): “showed more differences”

Comments 17: Line 263 - move sentence "Except for VSIG1 and ZIC2, all of the DEGs were also detected by StringTie soft (Table S2) after sentence in lines 265-268.

Response 17: Agree. We've moved that sentence to the back. You can see that in the re-submitted manuscript. Lines 273-274

Comments 18: Line 270 - replace "them located" by " them were located".

Response 18: Agree. You can see that in the re-submitted manuscript.

ž   Line 270 (Line 276): “all of them were located near two DEGs”

Comments 19: Line 326 - replace "to wild" by "with wild".

Response 19: Agree. You can see that in the re-submitted manuscript.

ž   Line 326 (Line 333): “Some breeds, for example Duroc, crossing with wild boar will produce piglets with the juvenile stripe.”

Comments 20: Line 359 - I am not sure if part of sentence "active or anagen at birth" is correct; maybe "active in anagen at birth".

Response 20: We didn't describe clearly before. At anagen period, hair follicles are active. Here, 95% of the hair follicles in the shoulders were active at birth, and then sharply decreased, with only about 50% of the anagen hair follicles after seven weeks. You can see that in the re-submitted manuscript.

ž   Line 359 (Line 366): “About 95% of the hair follicles were in anagen at birth, and then decreased to about 50% after seven weeks”

Comments 21: Lines 361-363 - explain more precisely terms "Postnatal P45" and "before 45d".

Response 21: "Postnatal P45" and "before 45d" both refer to the 45th day after birth. The primary hair follicles formed in the embryo begin to decay on the 45th day after birth.

ž   Lines 361-363 (Lines 368-369): “prenatal hair follicle morphogenesis development maintained until the 45th day after birth (45d)”

Comments 22: Line 382 - replace "In" by "Among".

Response 22: Agree. You can see that in the re-submitted manuscript.

ž   Line 382 (Line 389): “Among the 7 DE-miRNAs”

Comments 23: Line 382 - italicize gene symbol.

Response 23: Agree. You can see that in the re-submitted manuscript.

ž   Line 382 (Lines 389-290): “ssc-miR-532-3p probably participates in the formation”

Comments 24: Line 382 - replace "maybe" by "probably".

Response 24: Agree. You can see that in the re-submitted manuscript.

ž   Line 382 (Line 390): “ssc-miR-532-3p probably participates in the formation”

Comments 25: Line 410 - add underlined word "as found on chromosome 13, containing/covering two of 13"

Response 25: We understand that this was done to highlight the point and have made changes as you suggested.

Line 410 (Lines 417-418):” A hotspot of differentially expressing region between the light and dark stripes as found on chromosome 13, containing/covering two of 13.”

Comments 26: Lines 411-412 - italicize all genes symbols.

Response 26: Agree. You can see that in the re-submitted manuscript.

ž   Lines 411-412 (Lines 418-419): “A network among ZIC4 (DE-gene), ssc-miR-532-3p (DE-miRNA) and ENSSSCG00000056225 (DE-lncRNA) might regulate the juvenile stripe”

Comments 27: Table S2 - replace "Differently expression" by "Differently expressed".

Response 27: Agree. You can see that in the re-submitted supplementary files: Table S2 Differently expressed genes identified from featureCounts and StringTie

4. Response to Comments on the Quality of English Language

Point 1: Comments on the Quality of English Language :Section 2.3.2 must be corrected to give proper tense.

Response 1:   We appreciate your advice and apologize for the incorrect tense in the original manuscript. After becoming aware of this problem, we made changes to this paragraph. Lines 113-125.

Reviewer 2 Report

Comments and Suggestions for Authors

The paper addresses the issue related to the search for differentially expressed genes between the yellow and black stripes of the juvenile stripe in the offspring of wild boar and Duroc. The question is extremely interesting and relevant. In general, the offspring of wild boar and Duroc have the positive traits and meat quality of both wild boar and Duroc. Coat color is an important trait in pigs and can serve as a predictor for other breeding traits. The presence of juvenile stripes, which disappear as they grow older, is one of the characteristic features of the offspring of wild boar and Duroc. In this regard, the search for differentially expressed genes between the yellow and black stripes of the juvenile stripe in the offspring of wild boar and Duroc will reveal the mechanisms of this phenomenon.

The work plan, materials and methods are presented in detail in the article, correspond to the assigned tasks and, in general, do not raise any questions. All stages of the study are well presented, methods for determining the pigment content in hair, transcriptome analysis and data analysis are described.

Have a question regarding calculating threshold values ​​for differentially expressed gene scanning

1. How did you calculate the threshold values ​​for scanning differentially expressed genes (why did you choose |log2FoldChange| > 1)

There are also several questions based on the research results

 2. Based on the results of the work, differentially expressed mRNA (n=13), LncRNA (n=10) and microRNA (n= 4) were determined. Please clarify at what age this was established, since according to your method, samples were collected every 20 days.

3. In addition, what was the composition of pigments in the hair of light and dark stripes during the period of established differential expression? Can it be considered that certain mRNA (n=13), LncRNA (n=10) and microRNA (n= 4) are related in content pigments

4. The method refers to samples of Shanxia and Duroc pigs. At what stage are the results reported for these pigs, and has gene expression been studied in them?

Author Response

For review article

Response to Reviewer 2 Comments

1. Summary

Thank you very much for your attention and comments on our manuscript “Exploring the maintaining period and the differentially expressed genes between the yellow and black stripes of the juvenile stripe in the offspring of wild boar and Duroc”. We apologize for any confusion this may have caused you. Below, we will answer your questions point by point.

2. Questions for General Evaluation

Reviewer’s Evaluation

Response and Revisions

Does the introduction provide sufficient background and include all relevant references?

Yes/Can be improved/Must be improved/Not applicable

Is the research design appropriate?

Yes/Can be improved/Must be improved/Not applicable

Are the methods adequately described?

Yes/Can be improved/Must be improved/Not applicable

Are the results clearly presented?

Yes/Can be improved/Must be improved/Not applicable

Are the conclusions supported by the results?

Yes/Can be improved/Must be improved/Not applicable

3. Point-by-point response to Comments and Suggestions for Authors

Comments 1: The paper addresses the issue related to the search for differentially expressed genes between the yellow and black stripes of the juvenile stripe in the offspring of wild boar and Duroc. The question is extremely interesting and relevant. In general, the offspring of wild boar and Duroc have the positive traits and meat quality of both wild boar and Duroc. Coat color is an important trait in pigs and can serve as a predictor for other breeding traits. The presence of juvenile stripes, which disappear as they grow older, is one of the characteristic features of the offspring of wild boar and Duroc. In this regard, the search for differentially expressed genes between the yellow and black stripes of the juvenile stripe in the offspring of wild boar and Duroc will reveal the mechanisms of this phenomenon.

Response 1: Thank you for your review. As you said, we mainly explore why piglets of wild boar and Duroc have stripes by looking for genes that differ between stripes.

Comments 2: The work plan, materials and methods are presented in detail in the article, correspond to the assigned tasks and, in general, do not raise any questions. All stages of the study are well presented, methods for determining the pigment content in hair, transcriptome analysis and data analysis are described.

Response 2: Thank you for your recognition of our work.

Comments 3: Have a question regarding calculating threshold values for differentially expressed gene scanning.

Response 3: Your guiding suggestions will help us to further think about the scientific problems of the research.

Comments 4: How did you calculate the threshold values for scanning differentially expressed genes (why did you choose |log2FoldChange| > 1).

There are also several questions based on the research results.

Response 4: The threshold used in most transcriptome studies was |log2FoldChange| > 2 (there was a four-fold difference in gene expression between the case and control groups). What makes our study different from common studies is that the case and control groups had the same tissue type and were adjacent to each other. The difference between the two groups is small and the threshold needs to be lowered. In addition, we searched for literatures with similar conditions and found that the lowest threshold value was |log2FoldChange|=1 (the difference between groups was greater than 2). Therefore, we finally choose |log2FoldChange| > 1. We have added explanations to the text.

Line 169: “We used FDR < 0.05 and |log2FoldChange| > 1 as threshold values to scan the differentially expressed genes (DEGs) with similar groups.”

Comments 5: Based on the results of the work, differentially expressed mRNA (n=13), LncRNA (n=10) and microRNA (n= 4) were determined. Please clarify at what age this was established, since according to your method, samples were collected every 20 days.

Response 5: The samples we used for whole-transcriptome studies were skin tissues of approximately 30 days of age. This is mentioned in the methods section of this article. You can also see it on lines 97 and 136.

Line 97: “Five piglets were slaughtered at the age of 30 d, and the skin samples were collected from the dark and light stripes, respectively.”

Line 136: “Total RNAs were extracted from each skin sample”

Comments 6: In addition, what was the composition of pigments in the hair of light and dark stripes during the period of established differential expression?

Response 6: The pigments of the two stripes were eumelanin and pheomelanin. The color differences between stripes were mainly due to the differences in the content and distribution of the two pigments.

You can also see it on lines 49-50: ”During the implementation and maintenance periods, melanocytes in hair follicles produce melanin particles (eumelanin and pheomelanin)”.

Comments 7: Can it be considered that certain mRNA (n=13), lncRNA (n=10) and microRNA (n= 4) are related in content pigments.

Response 7: The functions of these differential expression genes may not be limited to pigment synthesis. Although we tried to control the differences between the two groups mainly for hair color, it was difficult to rule out other differences between the two groups, such as differences in skin thickness and hair length, and so on. Here, we reviewed the relevant literature and found that ESR1 can affect pigment synthesis and ZIC4 can participate in the formation of stripes. If we want to know whether these genes are related to pigment synthesis, we still need some experiments to verify it. Here, our work narrowed down the scope of verification through bioinformatics analysis. We have mentioned this in the discussion section (lines 382-391).

Comments 8: The method refers to samples of Shanxia and Duroc pigs. At what stage are the results reported for these pigs, and has gene expression been studied in them?

Response 8: Shanxia Black pig was bred as a paternal strain with high-quality meat and black coat color. The main feature of Duroc's appearance is its brownish red coat color. In the methods section, we solely collected hair from these two pig breeds (lines 92,93), and did not do transcriptomic analysis like WD to identify the expressed genes. The collected hair samples were used to determine ASM and EM content under different black hair ratios, and the results can be seen in figure S1 and section 3.2 (Pigment contents in the hairs of light and dark stripe).

Reviewer 3 Report

Comments and Suggestions for Authors

Page number

63

In this study, we crossed wild boars to Duroc

What is the need for crossing with Duroc; why Duroc chosen? Why not studied in wild boar itself

94

were slaughtered at the age

Why slaughtered why not collected the samples in live animals

325

Some breeds, for example Duroc, crossing to wild boars will produce piglets with the juvenile stripe.

Is stipes present in other breed cross; the line is misleading. If stripes are with some breeds only, then why not with other breeds

conclusion

A hotspot of differentially expressing region between the light and dark stripes was found on chromosome 13, two of 13 410 DE-genes and 8 of 10 DE-lncRNAs in this region.

How much accuracy we can say

Strength and weakness of the study is missing

Future direction of the results findings may be given

1. What is the main question addressed by the research?

Genetics of coat colour pattern in pigs

2. What parts do you consider original or relevant for the field? What specific gap in the field does the paper address?

Molecular genetic aspect

3. What does it add to the subject area compared with other published material?

A hotspot of differentially expressing region between the light and dark stripes was found on chromosome 13, two of 13  DE-genes and 8 of 10 DE-lncRNAs in this region.

4. What specific improvements should the authors consider regarding the
methodology? What further controls should be considered?.

Why this is studied in crossbred

5. Please describe how the conclusions are or are not consistent with the evidence and arguments presented. Please also indicate if all main questions posed were addressed and by which specific experiments

It addressed the objectives of the study and more precise highlights with future application may be given

6. Are the references appropriate?

Yes

Author Response

For review article

Response to Reviewer 3 Comments

1. Summary

Thank you very much for your attention and comments on our manuscript “Exploring the maintaining period and the differentially expressed genes between the yellow and black stripes of the juvenile stripe in the offspring of wild boar and Duroc”. Below, we will answer your questions point by point.

2. Questions for General Evaluation

Reviewer’s Evaluation

1. What is the main question addressed by the research?

Genetics of coat colour pattern in pigs

2. What parts do you consider original or relevant for the field? What specific gap in the field does the paper address?

Molecular genetic aspect

3. What does it add to the subject area compared with other published material?

A hotspot of differentially expressing region between the light and dark stripes was found on chromosome 13, two of 13 DE-genes and 8 of 10 DE-lncRNAs in this region.

4. What specific improvements should the authors consider regarding the
methodology? What further controls should be considered?

Why this is studied in crossbred

5. Please describe how the conclusions are or are not consistent with the evidence and arguments presented. Please also indicate if all main questions posed were addressed and by which specific experiments

It addressed the objectives of the study and more precise highlights with future application may be given

6. Are the references appropriate?

Yes

3. Point-by-point response to Comments and Suggestions for Authors

Comments 1: Page number 63; “In this study, we crossed wild boars to Duroc-What is the need for crossing with Duroc”; why Duroc chosen? Why not studied in wild boar itself

Response 1: Thank you for your questions. We have previously known that the offspring of a cross between a Duroc and a wild boar have juvenile stripes. Moreover, wild boars live in shrubs. 1) It is difficult to determine the time of delivery, unable to determine the age; 2) Difficulty in collecting hair samples at specific points in time; 3) Difficulty in obtaining skin samples. Therefore, we did not study the wild boar itself. We also added an explanation to the manuscript.

Line 63(Line 66): “Samples are difficult to obtain due to wild boar being wild animals”

Comments 2: Page number 94; “were slaughtered at the age”; Why slaughtered why not collected the samples in live animals.

Response 2: this is a meaningful question. The five piglets were born weak and malnourished. Samples are taken only after it has been determined that they are dead or beyond saving.

Comments 3: Page number 325; “Some breeds, for example Duroc, crossing to wild boars will produce piglets with the juvenile stripe”; Is stipes present in other breed cross; the line is misleading. If stripes are with some breeds only, then why not with other breeds.

Response 3: According to previous observation, the hybrid offspring of wild boar and Min pig also showed stripes, but they were not obvious. Additionally, some second-generation wild boars (wild boars crossed with domestic pigs) also exhibit distinct juvenile stripes, but we have not yet studied this and will further investigate this aspect in the future. In order to avoid misleading, we have made changes in the resubmitted manuscript.

Lines 332-333: “Crossing Duroc with wild boar will produce piglets with the juvenile stripe”

Comments 4: Conclusion; A hotspot of differentially expressing region between the light and dark stripes was found on chromosome 13, two of 13 410 DE-genes and 8 of 10 DE-lncRNAs in this region; How much accuracy we can say

Response 4: We cannot confirm until further research is conducted, but our current hypothesis suggests that members of the ZIC family may be involved in establishing the dorsal-ventral axis pattern during embryonic development. We anticipate that our future studies will bring clarity to this question.

Comments 5: Strength and weakness of the study is missing

Response 5: Thanks for your suggestions. I hope that my response below will provide insight into the strengths and weaknesses of this study. Strengths: (1) Although extensive analysis has been conducted on the pure coat color and phenotype of white belt with variable width in pigs, there remains a lack of understanding regarding more complex spatiotemporal patterns. Our research aims to address this gap and provide supplementary insights. (2) Additionally, we have provided a detailed description of the changes in stripes observed in striped pigs.

Weaknesses: (1) Research does have limitations, such as the challenge of determining the optimal time point for skin sampling. It would be beneficial to include additional time points for tracking hair color changes. (2) Additionally, the study is restricted to offspring of wild boar and Duroc, but more experimental populations can be included.

Comments 6: Future direction of the results findings may be given

Response 6: Your guiding suggestions will help us to further think about the research. We have several research directions for the current findings: (1) Extending the validation population. (2) Experiments will be conducted to verify that ZIC4 is involved in the development of striped pigs and whether it is regulated by nearby lncRNA. (3) Analyze the process of juvenile stripe at earlier developmental stage. (4) Consider epigenetic regulatory directions, such as ATAC-seq. (5) Ultimately, we want it to be able to use in the field of pet pig breeding. We also added it to the resubmitted manuscript.

Lines 411-413: “It is essential to continue determining the disappearance time of juvenile stripe. Additionally, we will conduct a comprehensive analysis, including validation of ZIC4 function and regulatory network, as well as epigenetic regulatory pathways.”